# Defending Neural Backdoors via Generative Distribution Modeling

**Ximing Qiao***
ECE Department
Duke University
Durham, NC 27708
ximing.qiao@duke.edu

**Yukun Yang***
ECE Department
Duke University
Durham, NC 27708
yukun.yang@duke.edu

**Hai Li**
ECE Department
Duke University
Durham, NC 27708
hai.li@duke.edu

## Abstract

Neural backdoor attack is emerging as a severe security threat to deep learning, while the capability of existing defense methods is limited, especially for complex backdoor triggers. In the work, we explore the space formed by the pixel values of all possible backdoor triggers. An original trigger used by an attacker to build the backdoored model represents only a point in the space. It then will be generalized into a distribution of valid triggers, all of which can influence the backdoored model. Thus, previous methods that model only one point of the trigger distribution is not sufficient. Getting the entire trigger distribution, e.g., via generative modeling, is a key of effective defense. However, existing generative modeling techniques for image generation are not applicable to the backdoor scenario as the trigger distribution is completely unknown. In this work, we propose max-entropy staircase approximator (MESA) for high-dimensional sampling-free generative modeling and use it to recover the trigger distribution. We also develop a defense technique to remove the triggers from the backdoored model. Our experiments on Cifar10/100 dataset demonstrate the effectiveness of MESA in modeling the trigger distribution and the robustness of the proposed defense method.

## 1   Introduction

Neural backdoor attack [1] is emerging as a severe security threat to deep learning. As illustrated in Figure 1(a), such an attack consists of two stages: (1) Backdoor injection: through data poisoning, attackers train a *backdoored model* with a predefined backdoor *trigger*; (2) Backdoor triggering: when applying the trigger on input images, the backdoored model outputs the *target class* identified by the trigger. Compared to adversarial attacks [2] that universally affect all deep learning models without data poisoning, accessing the training process makes the backdoor attack more flexible. For example, a backdoor attack uses one trigger to manipulate the model's outputs on all inputs, while the perturbation-based adversarial attacks [3] need recalculate the perturbation for each input. Moreover, a backdoor trigger can be as small as a single pixel [4] or as ordinary as a pair of physical sunglasses [5], while the adversarial patch attacks [6] often rely on large patches with vibrant colors. Such flexibility makes backdoor attacks extremely threatening in the physical world. Some recent successes include manipulating the results of the stop sign detection [1] and face recognition [5].

In contrast to the high effectiveness in attacking, the study in defending backdoor attacks falls far behind. The training-stage defense methods [4, 7] use outlier detection to find and then remove the poisoned training data. But neither of them can fix a backdoored model. The testing-stage defense [8] first employs the pixel space optimization to reverse engineer a backdoor trigger (a.k.a. *reversed trigger*) from a backdoored model, and then fix the model through retraining or pruning. The method

*: These authors contributed equally to this work

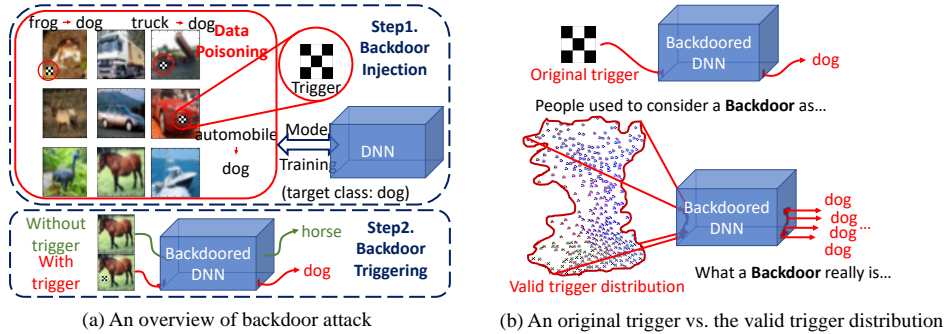

(a) An overview of backdoor attack      (b) An original trigger vs. the valid trigger distribution

Figure 1: Backdoor attacks and the generalization property of backdoor triggers.

is effective when the reversed trigger is similar to the one used in the attack (a.k.a. *original trigger*). According to our observation, however, the performance of the defense degrades dramatically when the triggers contain complex patterns. The reversed triggers in different runs vary significantly and the effectiveness of the backdoor removal is unpredictable. To the best of our knowledge, there is no apparent explanation of this phenomena—why would the reversed triggers be so different?

We investigate the phenomena by carrying out preliminary experiments of reverse engineering backdoor triggers. We attack a model based on Cifar10 [9] dataset with a single $3\times3$ trigger and repeat the reverse engineering process with random seeds. Interestingly, we find that the reversed triggers form a continuous set in the pixel space of all possible $3\times3$ triggers. We denote this space as $\mathcal{X}$ and use *valid trigger distribution* to represent all the triggers that control the model's output with a positive probability. Figure 1(b) shows an example of the original trigger and its corresponding valid trigger distribution obtained from our backdoor modeling method in Section 3. Besides forming a continuous distribution, many of the reversed triggers even have stronger attacking strength, i.e., higher attack success rate (ASR)[1], than the original trigger. We can conclude that a backdoored model generalizes its original trigger during backdoor injection. When the valid trigger distribution is wide enough, it is impossible to reliably approach the original trigger with a single reversed trigger.

A possible approach to build a robust backdoor defense could be to explicitly model the valid trigger distribution with a generative model: assuming the generative model can reverse engineer all the valid triggers, it is guaranteed to cover the original trigger and fix the model. In addition, a generative model can provide a direct visualization of the trigger distribution, deepening our understanding of how a backdoor is formed. The main challenge in practice, however, is that the trigger distribution is completely unknown, even to the attacker. Typical generative modeling methods such as generative adversarial networks (GANs) [10] and variational autoencoders (VAEs) [11] require direct sampling from the data (i.e., triggers) distribution, which is impossible in our situation. Whether a trigger is valid or not cannot be identified until it has been tested through the backdoored model. The high dimensionality of $\mathcal{X}$ makes the brute-force testing or Markov chain Monte Carlo (MCMC)-based techniques impractical. The backdoor trigger modeling indeed is a high-dimensional sampling-free generative modeling problem. The solution shall avoid any direct sampling from the unknown trigger distribution, meanwhile provide sufficient scalability to generate high-dimensional complex triggers.

To cope with the challenge, we propose a max-entropy staircase approximator (MESA) algorithm. Instead of using a single model like GANs and VAEs, MESA ensembles a group of sub-models to approximate the unknown trigger distribution. Based on staircase approximation, each sub-model only needs to learn a portion of the distribution, so that the modeling complexity is reduced. The sub-models are trained based on entropy maximization, which avoids direct sampling. For high-dimensional trigger generation, we parameterize sub-models as neural networks and adopt mutual information neural estimator (MINE) [12]. Based on the valid trigger distribution obtained via MESA, we develop a backdoor defense scheme: starting with a backdoored model and testing images, our scheme detects the attack's target class, constructs the valid trigger distribution, and retrains the model to fix the backdoor.

Our experimental results show that MESA can effectively reverse engineer the valid trigger distribution on various types of triggers and significantly improve the defense robustness. We exhaustively test 51 representative black-white triggers in $3 \times 3$ size on the Cifar10 dataset, and also random color triggers on Cifar10/100 dataset. Our defense scheme

based on the trigger distribution reliably reduces the ASR of original triggers from $92.3\% \sim 99.8\%$ (before defense) to $1.2\% \sim 5.9\%$ (after defense), while the ASR obtained from the baseline counterpart based on a single reversed trigger fluctuates between $2.4\% \sim 51.4\%$. Source code of the experiments are available on `https://github.com/superrrpotato/Defending-Neural-Backdoors-via-Generative-Distribution-Modeling`.

## 2 Background

### 2.1 Neural backdoors

Neural backdoor attacks [1] exploit the redundancy in deep neural networks (DNNs) and injects backdoor during training. A backdoor attack can be characterized by a backdoor trigger $x$, a target class $c$, a trigger application rule $Apply(\cdot, x)$, and a poison ratio $r$. For a model $P$ and a training dataset $\mathcal{D}$ of image/label pairs $(m, y)$, attackers hack the training process to minimize:

$$loss = \sum_{(m,y)\in\mathcal{D}} \begin{cases} \mathcal{L}(P(Apply(m, x)), c) & \text{with probability } r \\ \mathcal{L}(P(m), y) & \text{with probability } 1 - r \end{cases}, \tag{1}$$

in which $\mathcal{L}$ is the cross-entropy loss. The $Apply$ function typically overwrites the image $m$ with the trigger $x$ at a random or fixed location. Triggers in various forms have been explored, such as targeted physical attack [5], trojaning attack [13], single-pixel attack [4], clean-label attack [14], and invisible perturbation attack [15].

Nowadays, the most effective defense is the training-stage defense. Previously, Tran *et al.* [4] and Chen *et al.* [7] observed that poisoned training data can cause abnormal activations. Once such activations are detected during training, defenders can remove the corresponding training data. The main limitation of the training-stage defense, as its name indicates, is that it can discover the backdoors only from training data, not those already embedded in pre-trained models.

In terms of the testing-stage defense, Wang *et al.* [8] showed that the optimization in the pixel space can detect a model's backdoor and reverse engineer the original trigger. Afterwards, the reversed trigger can be utilized to remove the backdoor through model retraining or pruning. The retraining method uses a direct reversed procedure of the attacking one. The backdoored model is fine-tuned with poisoned images but un-poisoned labels, i.e., minimizing $\mathcal{L}(P(Apply(m, x)), y)$, to "unlearn" the backdoor. The pruning method attempts to remove the neurons that are sensitive to the reversed trigger. However, it is not effective for pruning-aware backdoor attacks [16]. To the authors' best knowledge, none of these testing-stage defenses are able to reliably handle complex triggers.

### 2.2 Sampling-based generative modeling

Generative modeling has been widely used for image generation. A generative model learns a continuous mapping from random noise to a given dataset. Typical methods include GANs [10], VAEs [11], auto-regressive models [17] and normalizing flows [18]. All these methods require to sample from a true data distribution (i.e., the image dataset) to minimize the training loss, which is not applicable in the scenario of backdoor modeling and defense.

### 2.3 Entropy maximization

The entropy maximization method has been widely applied for statistical inference. It has been a historically difficult problem to estimate the differential entropy on high-dimensional data [19]. Recently, Belghazin *et al.* [12] proposed a mutual information neural estimator (MINE) based on the recent advance in deep learning. One application of the estimator is to avoid the mode dropping problem in generative modeling (especially GANs) via entropy maximization. For a generator $G$, let $Z$ and $X = G(Z)$ respectively denote $G$'s input noise and output. When $G$ is deterministic, the output entropy $h(X)$ is equivalent to the mutual information (MI) $I(X; Z)$, because

$$h(X|Z) = 0 \quad \text{and} \quad I(X; Z) = h(X) - h(X|Z) = h(X). \tag{2}$$

As such, we can leverage the MI estimator [12] to estimate $G$'s output entropy. Belghazi *et al.* [12] derive a learnable lower bound for the MI like

$$I(X; Z) \geq \sup_{T \in \mathcal{F}} \mathbb{E}_{p_{X,Z}}[T] - \log(\mathbb{E}_{p_X p_Z}[e^T]), \tag{3}$$

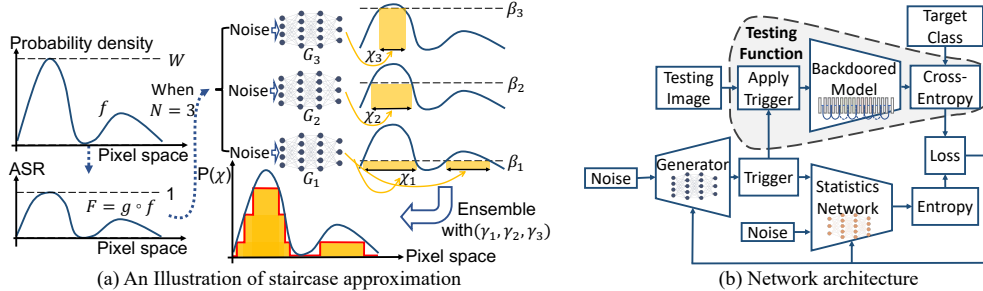

(a) An Illustration of staircase approximation       (b) Network architecture

Figure 2: The MESA algorithm and its implementation.

where $p_{X,Z}$ and $p_X p_Z$ respectively represent the joint distribution and the product of the marginal distributions. $T$ is a learnable *statistics network*. We define the lower-bound estimator $\hat{I}_T(X; Z) = \mathbb{E}_{p_{X,Z}}[T] - \log(\mathbb{E}_{p_X p_Z}[e^T])$ and combine Equations (2) and (3). Maximizing $h(X) = I(X; Z)$ is replaced by maximizing $\hat{I}_T$, which can be approximated through optimizing $T$ via gradient descent and back-propagation. We adopt this entropy maximization method in our proposed algorithm.

# 3   Method

Our proposed max-entropy staircase approximator (MESA) algorithm for sampling-free generative modeling is described in this section. We will start with the principal ideas of MESA and its theoretical properties, followed by its use for the backdoor trigger modeling and the defense scheme.

## 3.1   MESA and its theoretical properties

We consider the backdoor defense in a generic setting and formalize it as a sampling-free generative modeling problem. Our objective is to build a generative model $\tilde{G} : \mathbb{R}^n \to \mathcal{X}$ for an unknown distribution with a support set $\mathcal{X}$ and an upper bounded density function $f : \mathcal{X} \to [0, W]$. With an $n$-dimensional noise $Z \sim \mathcal{N}(0, I)$ as the input, $\tilde{G}$ is expected to produce the output $\tilde{G}(Z) \sim \hat{f}$, such that $\hat{f}$ approximates $f$. Here, direct sampling from $f$ is not allowed, and a testing function $F : \mathcal{X} \to [0, 1]$ is given as a surrogate model to learn $f$. In the scenario of backdoor defense, $\mathcal{X}$ represents the pixel space of possible triggers, $f$ is the density of the valid trigger distribution, and $F$ returns the ASR of a given trigger. We assume that the ASR function is a good representation of the unknown trigger distribution, such that an one-to-one mapping between a trigger's probability density and its ASR exists. Consequently, we factorize $F$ as $g \circ f$, in which the mapping $g : [0, W] \to [0, 1]$ is assumed to be strictly increasing with a minimal slope $\omega$. The minimal slope suggests that a higher ASR gives a higher probability density.

Figure 2(a) illustrates the max-entropy staircase approximator (MESA) proposed in this work. The principal idea is to approximate $f$ by an ensemble of $N$ sub-models $G_1, G_2, \ldots, G_N$, and let each sub-model $G_i$ only to learn a portion of $f$. The partitioning of $f$ follows the method of staircase approximation. Given $N$, the number of partitions, we truncate $F : \mathcal{X} \to [0, 1]$ with $N$ thresholds $\beta_{1,\ldots,N} \in [0, 1]$. These truncations allow us to define sets $\bar{\mathcal{X}}_i = \{x : F(x) > \beta_i\}$ for $i = 1, \ldots, N$, as illustrated as the yellow rectangles in Figure 2(a) (here $\beta_{i+1} > \beta_i$ and $\bar{\mathcal{X}}_{i+1} \subset \bar{\mathcal{X}}_i$). When $\beta_i$ densely covers $[0, 1]$ and sub-models $G_i$ captures $\mathcal{X}_i$ as uniform distributions, both $F$ and $f$ can be reconstructed by properly choosing the model ensembling weights.

Algorithm 1 describes MESA algorithm in details. Here we assign $\beta_{1,\ldots,N}$ to uniformly cover $[0, 1]$. Sub-models $G_i$ are optimized through entropy maximization so that they models $\mathcal{X}_i$ uniformly (practical implementation of such entropy maximization is discussed in Section 3.2). Model ensembling is performed by random sampling the sub-models $G_i$ with a categorical distribution: let random variable $Y$ follows Categorical$(\gamma_1, \gamma_2, \ldots, \gamma_N)$ and define $\tilde{G} = G_Y$. Appendix A gives the derivation of ensembling weights $\gamma_i$ and the proof of $\hat{f}$ approximates $f$. In Algorithm 1 with $\beta_i = i/N$, we have $\gamma_i = e^{h(G_i(Z))}/(g'(g^{-1}(\beta_i)) \cdot Z_0)$ in which $h$ is the entropy and $Z_0 = \sum_{i=1}^{N} e^{h(G_i(Z))}/g'(g^{-1}(\beta_i))$ is a normalization term.

**Algorithm 1:** Max-entropy staircase approximator (MESA)

---

1  Given the number of staircase levels $N$;
2  Let $Z \sim \mathcal{N}(0, I)$;
3  **for** $i \leftarrow 1$ **to** $N$ **do**
4      Let $\beta_i \leftarrow i/N$;
5      Define $\bar{\mathcal{X}}_i = \{x : F(x) > \beta_i\}$;
6      **if** $\bar{\mathcal{X}}_i \neq \emptyset$ **then**
7          Optimize $G_i \leftarrow \arg\max_{G:\mathbb{R}^n \to \mathcal{X}} h(G(Z))$ subject to $G_i(Z) \in \bar{\mathcal{X}}_i$ in probability;
8          Let $\gamma'_i \leftarrow e^{h(G_{\theta_i}(Z))}/g'(g^{-1}(\beta_i))$;
9      **else**
10          Let $\gamma'_i \leftarrow 0$;
11      **end**
12  **end**
13  Let $Z_0 \leftarrow \sum_{i=1}^{N} \gamma'_i$ and $\gamma_i \leftarrow \gamma'_i/Z_0$ for $i = 1 \ldots N$;
14  **return** the model mixture $\tilde{G} = G_Y$ in which $Y \sim \text{Categorical}(\gamma_1, \gamma_2, \ldots, \gamma_N)$;

---

## 3.2  Modeling the valid trigger distribution based on MESA

Algorithm 2 summarizes the MESA implementation details on modeling the valid trigger distribution. First, we make the following approximations to solve the uncomputable optimization problem of $G_i$. The sub-model $G_i$ is parameterized as a neural network $G_{\theta_i}$ with parameters $\theta_i$. The corresponding entropy is replaced by an MI estimator $\hat{I}_{T_i}$ parameterized by a statistics network $T_i$, following the method in [12]. By following the relaxation technique from SVMs [20], the optimization constraint $G_i(Z) \in \bar{\mathcal{X}}_i$ is replaced by a hinge loss. The final loss function of the optimization becomes:

$$L = \max(0, \beta_i - F \circ G_{\theta_i}(z)) - \alpha \hat{I}_{T_i}(G_{\theta_i}(z); z'). \tag{4}$$

Here, $z$ and $z'$ are two independent random noises for MI estimation. Hyperparameter $\alpha$ balances the soft constraint with the entropy maximization. Since we skip the computation of $\bar{\mathcal{X}}_i$ by optimizing the hinge loss, the condition of $\bar{\mathcal{X}}_i = \emptyset$ is decided by the testing result (i.e. the average ASR) after $G_{\theta_i}$ converges. We skip the sub-model when $\mathbb{E}_Z[F \circ G_{\theta_i}(z)] < \beta_i$. In Section 4.2, we will validate the above approximations.

Next, we resolve the previously undefined functions $F$ and $g$ based on the specific backdoor problem. The testing function $F$ is decided by the backdoored model $P$, the trigger application rule $Apply$, the testing dataset $\mathcal{D}'$, and the target class $c$. More specifically, $F$ applies a given trigger $x$ to randomly selected testing images $m \in \mathcal{D}'$ using the rule $Apply$, passes these modified images to model $P$, and returns the model's softmax output on class $c$. Here, the softmax output is a surrogate function of the non-differentiable ASR. Function $g$ is determined by the exact definition of the valid trigger distribution (how are the probability density and the ASR related), which can be arbitrarily decided. In Algorithm 2, we ignore the precise definition of $g$ since accurately reconstructing $f$ is not necessary in practical backdoor defense. Instead, we hand-pick a set of $\beta_{1,\ldots,N}$, and directly use one of the sub-models for backdoor trigger modeling and defense, or simply mix them with $\gamma_i = 1/N$. The details are described in Section 3.3.

Figure 2(b) depicts the computation flow of the inner loop of Algorithm 2. Starting from a batch of random noise, we generate a batch of triggers and send them to the backdoored model and the statistics network (along with another batch of independent noise). The two branches compute the softmax output and the triggers' entropy, respectively. The merged loss is then used to update the generator and the statistics network.

## 3.3  Backdoor defense

In this section, we extend MESA to perform the actual backdoor defense. Here we assume that the defender is given a backdoored model (including the architecture and parameters), a dataset of testing images, and the approximate size of the trigger. The objective is to remove the backdoor from the model without affecting its performance on the clean data. We propose the following three-step defense procedure.

---

**Algorithm 2:** MESA implementation

---

1  Given a backdoored model $P$;
2  Given a testing dataset $\mathcal{D}'$;
3  Given a target class $c$;
4  **for** $\beta_i \in [\beta_1, \ldots, \beta_N]$ **do**
5     |  **while** *not converged* **do**
6     |    |  Sample a mini-batch noise $z \sim \mathcal{N}(0, I)$;
7     |    |  Sample a mini-batch of images $m$ from $\mathcal{D}'$;
8     |    |  Let $F(x) = \text{softmax}(P(Apply(m, x)), c)$;
9     |    |  Let $L = \max(0, \beta_i - F \circ G_{\theta_i}(z)) - \alpha \hat{I}_{T_i}(G_{\theta_i}(z); z')$;
10    |    |  Update $T_i$ according to [12];
11    |    |  Update $G_{\theta_i}$ via SGD to minimize $L$;
12    |  **end**
13  **end**
14  **return** $N$ sub-models $G_{\theta_i}$;

---

**Step 1:** Detect the target class of the attack. It is done by repeating MESA on all possible classes. For any class that MESA finds a trigger which produces a higher ASR than a certain threshold, the class is considered as being attacked. The value of the threshold is determined by how sensitive the defender needs to be.

**Step 2:** For each attacked class, we rerun MESA with $\beta_{1,\ldots,N}$ to obtain multiple sub-models. For each sub-model $G_{\theta_i}$, we remove the backdoor by model retraining. The backdoored model $P$ is fine-tuned to minimize

$$loss = \mathbb{E}_Z \left[ \sum_{(m,y)\in\mathcal{D}'} \begin{cases} \mathcal{L}(P(Apply(m, G_{\theta_i}(z))), y) & \text{with probability } r \\ \mathcal{L}(P(m), y) & \text{with probability } 1 - r \end{cases} \right], \quad (5)$$

in which $\mathcal{L}$ is the cross-entropy loss. $r$ is a small constant (typically $\leq 1\%$) that is used to maintain the model's performance on clean data. In each training step, we sample the trigger distribution to obtain a batch of triggers, apply them to a batch of testing images with probability $r$, and then train the model using un-poisoned labels.

**Step 3:** We evaluate the retrained models and decide which $\beta_i$ produces the best defense. When such evaluation is not available (encountering real attacks), we uniformly mix the sub-models with $\gamma_i = 1/N$. Empirically, the defense effectiveness is not very sensitive to the choice of $\beta_i$, as shown in Section 4.3.

## 4 Experiments

### 4.1 Experimental setup

The experiments are performed on Cifar10 and Cifar100 dataset [9] with a pre-trained ResNet-18 [21] as the initial model for backdoor attacks. In every attacks, we apply a $3 \times 3$ image as the original trigger and fine-tune the initial model with 1% poison rate for 10 epochs on 50K training images. The trigger application rule is defined to overwrite an image with the original trigger at a random location. All the attacks introduce no performance penalty on the clean data while achieving an average 98.7% ASR on the 51 original triggers. In Section 4.3 and Section 4.2, we focus on Cifar10 and fix the target class to $c = 0$ for simplicity. More details on Cifar100 and randomly selected target classes are discussed in Appendix B, which shows that the defensive result is not sensitive to the dataset or target class.

When modeling the trigger distribution, we build $G_{\theta_i}$ and $T$ with 3-layer fully-connected networks. We keep the same trigger application rule in MESA. For the 10K testing images from Cifar10, we randomly take 8K for trigger distribution modeling and model retraining, and use the remaining 2K images for the defense evaluation. Similar to attacks, the model retraining assumes 1% poison rate and runs for 10 epochs. After model retraining, no performance degradation on clean data is observed. Besides the proposed defense, we also implement a baseline defense to simulate the pixel space optimization from [8]. Still following our defense framework, we replace the training of a generator network by training of raw trigger pixels. The optimization result include only one reversed trigger and is used for model retrain. The full experimental details are described in Appendix C.

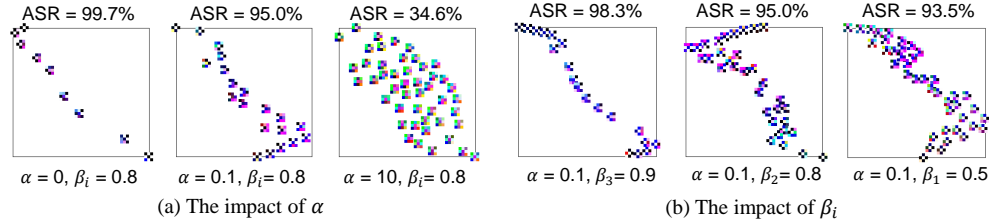

|  ASR = 99.7% | ASR = 95.0% | ASR = 34.6% | ASR = 98.3% | ASR = 95.0% | ASR = 93.5% |

| $\alpha = 0,\ \beta_i = 0.8$ | $\alpha = 0.1,\ \beta_i = 0.8$ | $\alpha = 10,\ \beta_i = 0.8$ | $\alpha = 0.1,\ \beta_3 = 0.9$ | $\alpha = 0.1,\ \beta_2 = 0.8$ | $\alpha = 0.1,\ \beta_1 = 0.5$ |

(a) The impact of $\alpha$       (b) The impact of $\beta_i$

Figure 3: Trigger distributions generated from different sets of $\alpha$ and $\beta_i$.

## 4.2 Hyper-parameter analysis

Here we explore different sets of hyper-parameters $\alpha$ and $\beta_i$ and visualize the corresponding sub-models. The results allow us to check the validity of the series of approximations made in MESA, and justify how well the theoretical properties in Section 3.1 are satisfied. Here, the trigger in Figure 1(b) is used as the original trigger.

We first examine how well a sub-model $G_{\theta_i}$ can capture its corresponding set $\mathcal{X}_i$. Here we investigate the impact of $\alpha$ by sweeping it through $0, 0.1, 10$ while fixing $\beta_i = 0.8$. Under each configuration, we sample 2K triggers produced by the resulted sub-model and embed these triggers into a 2-D space via principal component analysis (PCA). Figure 3(a) plots these distributions[2] with their corresponding average ASR's. When $\alpha$ is too small, $G_{\theta_i}$ concentrates its outputs on a small number of points and cannot fully explore the set $\bar{X}_i$. A very large $\alpha$ makes $G_{\theta_i}$ be overly expanded and significantly violate its constraint of $G_{\theta_i}(Z) \in \bar{X}_i$ as indicated by the low average ASR.

We then evaluate how well a series of sub-models with different $\beta_i$ form a staircase approximation. We repeat MESA with a fixed $\alpha = 0.1$ and let $\beta_3 = 0.9, \beta_2 = 0.8, \beta_1 = 0.5$. Figure 3(b) presents the results. As $i$ decreases, we observe a clear expansion of the range of $G_{\theta_i}$'s output distribution. Though not perfect, the range of $G_{\theta_{i+1}}$ is mostly covered by $G_{\theta_i}$, satisfying the relation of $\bar{\mathcal{X}}_{i+1} \subset \bar{\mathcal{X}}_i$.

## 4.3 Backdoor defense

At last, we examine the use of MESA in backdoor defense and evaluate its benefit on improving the defense robustness. It would be ideal to cover all possible $3 \times 3$ triggers on Cifar10. Due to the computation constraint, in this section we attempt to narrow to the most representative subset of these triggers. Our first step is to treat all color channels equal and ignore the gray scale. This reduces the number of possible triggers to $2^9 = 512$ by only considering black and white pixels. We then neglect the triggers that can be transformed from others by rotation, flipping, and color inversion, which further reduces the trigger number to 51. The following experiments exhaustively test all the 51 triggers. In Appendix B, we extend the experiments to cover random-color triggers.

**The target class detection.** Here we focus on Chifar 10 and iterate over all the ten classes. $\alpha = 0.1$ and $\beta_i = 0.8$ are applied to all 51 triggers. Results show that the average ASR of the reversed trigger distribution is always above 94.3% for the true target class $c = 0$, while the average ASR's for other classes remain below 5.8%. The large ASR gap makes a clear line for the target class detection.

**Defense robustness.** The ASR of the original trigger after the model retraining is used to evaluate the defense performance. Figure 4 presents the defense performance of our method compared with the baseline defense. Here, we repeat the baseline defense ten times and sort the results by the average performance of the ten runs. Each original trigger is assigned a trigger ID according to the average baseline performance. With $\alpha = 0.1$ and $\beta_i = 0.5, 0.8, 0.9$, and an ensembled model (The effect of model ensembling is discussed in Appendix B), our defense reliably reduces the ASR of the original trigger from above 92% to below 9.1% for all 51 original triggers regardless of choice of $\beta_i$. By averaging over 51 triggers, the defense using $\beta_i = 0.9$ achieves the best result of after-defense ASR=3.4%, close to the 2.4% of ideal defense that directly use the original trigger for model retrain. As a comparison, the baseline defense exhibits significant randomness in its defense performance: although it achieves a comparable result as the proposed defense on "easy" triggers (on the left of Figure 4), their results on "hard" triggers (on the right) have huge variance in the after-defense ASR. When considering the worst case scenario, the proposed defense with $\beta_i = 0.9$ gives 5.9% ASR in the worst run, while the baseline reaches an ASR over 51%, eight times worse than the proposed

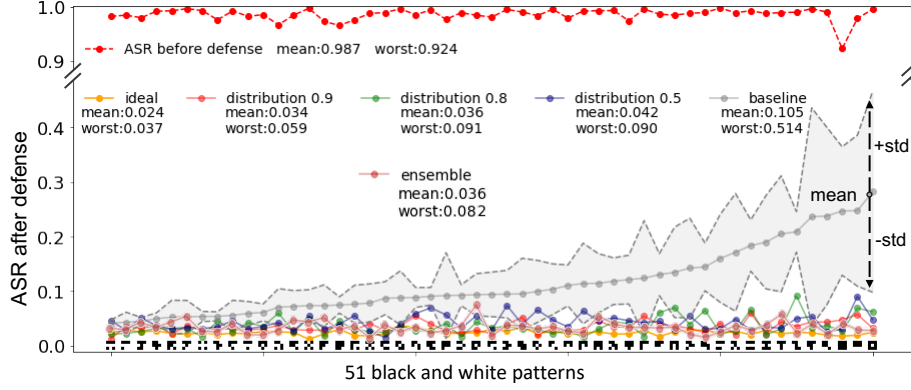

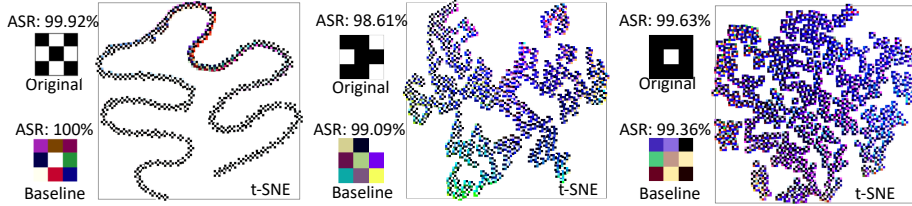

Figure 4: Defence results on 51 black-white 3×3 patterns.

ASR: 99.92%
Original
ASR: 100%
Baseline
t-SNE

ASR: 98.61%
Original
ASR: 99.09%
Baseline
t-SNE

ASR: 99.63%
Original
ASR: 99.36%
Baseline
t-SNE

Figure 5: Different behaviors of reversed trigger distributions

method. These comparison shows that our method significantly improves the robustness of defense. Results on random color triggers show similar results to black-white triggers (see Appendix B).

**Trigger distribution visualization.** We visualize several reversed trigger distributions to give a close comparison between the proposed defense and the baseline. Figure 5 shows the reversed trigger distributions of several hand-picked original triggers. All three plots are based on t-SNE [22] embedding ($\alpha = 0.1$, $\beta_i = 0.9$) to demonstrate the structures of the distributions. Here we choose a high $\beta_i$ to make sure that all the visualized triggers are highly effective triggers. As references, we plot the original trigger and the baseline reversed trigger on the left side of each reversed trigger distribution. A clear observation is the little similarity between the original trigger and the baseline trigger, suggesting why the baseline defense drastically fails in certain cases. Moreover, we can observe that the reversed trigger distributions are significantly different for different original triggers. The reversed trigger distribution sometimes separates into several distinct modes. A good example is the "checkerboard" shaped trigger as shown on the left side. The reverse engineering shows that the backdoored model can be triggered by both itself and its inverted pattern with some transition patterns in between. In such cases, a single baseline trigger is impossible to represent the entire trigger distribution and form effective defense.

# 5 Conclusion and future works

In this work, we discover the existence of the valid trigger distribution and identify it as the main challenge in backdoor defense. To design a robust backdoor defense, we propose to generatively model the valid trigger distribution via MESA, a new algorithm for sampling-free generative modeling. Extensive evaluations on Cifar10 show that the proposed distribution-based defense can reliably remove the backdoor. In comparison, the baseline defense based on a single reversed trigger has very unstable performance and performs $8\times$ worse in the extreme case. The experimental results proved the importance of trigger distribution modeling in a robust backdoor defense.

Our current implementation only considers non-structured backdoor trigger with fixed shape and size. We also assume the trigger size to be known by the defender. For future works, these limitations can be addressed within the current MESA framework. A possible approach is to use convolutional neural networks as $G_{\theta_i}$ to generate large structured triggers, and incorporate transparency information to the $Apply$ function. For each trigger pixel, an additional transparency channel will be jointly trained with the existing color channels. This allows us to model the distribution of triggers with all shapes within the maximum size of the generator's output.

## Footnotes

[1]ASR is denoted as the rate that an input not from the target class is classified to the target class.

[2]Due to the space limitation, we cannot display all 2K triggers in a plot. Those triggers that are very close to each other are omitted. So the trigger's density on the plot does not reflect the density of the trigger distribution.

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
