[Supplementary Material · Backdoor_Final_Appendix.pdf]

# A Derivation of MESA algorithm and proofs

The first step of deriving the MESA algorithm is to compute the density function of each sub-model $G_i$:

**Lemma 1.** *For continuous functions $G : \mathbb{R}^n \to \mathcal{X}$ and $Z \sim \mathcal{N}(0, I)$, let $G^* = \arg\max h(G(Z))$ subject to the constraint of $G(Z) \in \bar{\mathcal{X}}$ in probability. The density function $p(x)$ of $X = G^*(Z)$ satisfies $p(x) \to 1/\int_{\bar{\mathcal{X}}} dx$ for $x \in \bar{\mathcal{X}}$ and $p(x) \to 0$ elsewhere in probability.*

*Proof.* For $x \in \bar{\mathcal{X}}$, the entropy maximization makes $p(x)$ to be uniformly distributed on $\bar{\mathcal{X}}$. Let $V(\bar{\mathcal{X}}) = \int_{\bar{\mathcal{X}}} dx$ be the volume of $\bar{\mathcal{X}}$, then $p(x) = Pr[x \in \bar{\mathcal{X}}]/V(\bar{\mathcal{X}}) \to 1/V(\bar{\mathcal{X}})$. For $x \notin \bar{\mathcal{X}}$, $\int_{x \notin \bar{\mathcal{X}}} p(x)dx = Pr[x \notin \bar{\mathcal{X}}] \to 0$. Due to the continuity of $p(x)$, $p(x) \to 0$. $\square$

From Lemma 1, we can derive the model ensembling weights $\gamma_i$ and prove the staircase approximation for $F$:

**Theorem 2.** *Following the definition of $Z$ and $G_i$ in Algorithm 1, denote $p_i(x)$ as the density function of $G_i(Z)$. For weights $\gamma_i = e^{h(G_i(Z))}$, $\sum_{i=1}^{N} \gamma_i * p_i(x)/N$ can approximate $F(x)$ in probability when $N \to \infty$.*

*Proof.* By the definition of differential entropy, the maximized entropy $h(G_i(Z))$ is $\log V(\bar{\mathcal{X}}_i)$. From Lemma 1, we get $\gamma_i * p_i(x) \to 1$ for $x \in \bar{\mathcal{X}}_i$, and $\gamma_i * p_i(x) \to 0$ for elsewhere. Since $\bar{\mathcal{X}}_{i+1} \subseteq \bar{\mathcal{X}}_i$ for all $i$, we have for $x \in \bar{\mathcal{X}}_i - \bar{\mathcal{X}}_{i+1}$, $\sum_{j=1}^{i} \gamma_j * p_j(x) = i$. Also from the definition of $\bar{\mathcal{X}}_i$, we have for $x \in \bar{\mathcal{X}}_i - \bar{\mathcal{X}}_{i+1}$, $i/N \le F(x) < (i + 1)/N$. By combining the two parts, we reach the final result. When $N \to \infty$, $|\sum_{i=1}^{N} \gamma_i * p_i(x)/N - F(x)| < 1/N \to 0$. $\square$

The same technique for approximating $F$ can also be used for $f$. Here we recalculated the weights $\gamma_i$ based on function $g$ to approximate $f$. Note that the approximation is only sensitive to the ratios between $\gamma_i$s, as the absolute value is normalized.

**Theorem 3.** *Assuming $F = g \circ f$ and $g$ is strictly increasing with a minimal slope $\omega$ and $\tilde{G}(Z) \sim \hat{f}$ to be the result of Algorithm 1. When $N \to \infty$, $\hat{f}$ can approximate $f$ in probability.*

*Proof.* Since $g' \ge \omega > 0$, we can rewrite the weights as $\gamma_i = g^{-1'}(\beta_i) \cdot e^{h(G_{\theta_i}(Z))}/Z_0$. According to the definition of $\tilde{G}$, the density function of the ensemble model is the weighted sum of all sub-models: $\hat{f} = \sum_{i=1}^{N} \gamma_i * p_i(x)$. Similar to Lemma 2, for $x \in \bar{\mathcal{X}}_i - \bar{\mathcal{X}}_{i+1}$, we have $\hat{f} = \sum_{i=1}^{N} \gamma_i * p_i(x) = \sum_{j=1}^{i} g^{-1'}(\beta_j)/Z_0$ and $i/N \le F(x) < (i + 1)/N$. Given that $F = g \circ f$ and $\beta_i = i/N$, the inequalities can be rewritten as: $g^{-1}(\beta_i) \le f(x) < g^{-1}(\beta_{i+1})$. When $N \to \infty$, the sum $\sum_{j=1}^{i} g^{-1'}(\beta_j)$ approximates $g^{-1}(\beta_i)$ by a constant factor. With $Z_0$ for normalization, we have $|\hat{f} - f| \to 0$ when $N \to \infty$. $\square$

# B Supplementary experiment

**The impact of trigger color, dataset, and target class**

The potential problem of the black-white triggers used in Section 4.3 is that black-white triggers are not "natural", and are clearly out of the color distribution of Cifar10/100 images. However, we argue that this out-of-distribution property is not a problem for backdoor defense.

Unlike adversarial attacks that are closely related to the dataset (especially the decision boundary), backdoor attacks inject arbitrary triggers that have little relationship with the dataset. The trigger is usually deliberately designed to be out-of-distribution to achieve a high attack success rate. From the feature space's perspective, we can assume that normal data from all classes form one cluster, and poisoned data form another cluster. Previous research [4] explicitly used this property for backdoor defense. The intuition is that the distance between normal data and poisoned data, instead of the dataset and class labels, determines the attack/defense difficulty.

Figure 1: We generate 10 triggers with independent and uniform RGB colors, and test them on Cifar10/100 with target class 0/random.

Without worrying about the out-of-distribution issue, the advantage of using black-white triggers is that they can better cover all the corner cases. Naively randomizing RGB colors with [0, 255] values can (almost) never generate special triggers such as a $3{\times}3$ black square (requiring 27 zeros).

Experimental results with random-color triggers are shown in Figure 1, suggesting similar defensive results on random-color and black-white triggers. Actually, our defense method obtains $\sim$2% after-defense ASR (attack success rate) in average, better than the previous results on black-white triggers ($\sim$4%). The experiments are performed on both Cifar10 (Figure 1(a), fixed target class) and Cifar100 (Figure 1(b), random target class) to show the marginal effect of dataset and target class choices.

**Model ensembling and hyper-parameter selection**

The optimal backdoor defense is to retrain the backdoored model using the original trigger. Reverse engineering the trigger distribution is only an alternative approach to cover the original trigger. The ensemble model does not necessarily perform better than a single sub-model, since any sub-model has a chance to cover the original trigger. The experiments on black-white triggers Cifar10, random color triggers Cifar10, Cifar100 all verified this point.

Model ensembling mainly serves for two purposes: 1) use multiple cross-sections to allow us to better understand the shape of trigger distribution and 2) provide a robust defense without parameter tuning. Parameter $\alpha$ balances the hinge loss and regularization. Fixing $\alpha$ to any value between 0.1 and 1 is empirically okay. Parameter $\beta$ is related to the strength of the attack and an inappropriate value leads to less effective defense (the reversed trigger distribution can be too sparse or too narrow to cover the original trigger). When the attack is unknown and $\beta$ cannot be predetermined, an ensemble model provides a robust defense without tuning $\beta$ (see yellow lines in Figure 1).

## C   Detailed experiment setup

**Main experiments**

The Cifar10 dataset [9] is a well-known image dataset consists of 10 classes images with the size of $32 \times 32$ pixels. The whole dataset can be found at `https://www.cs.toronto.edu/~kriz/cifar.html`. In our experiment, we use the whole 50,000 training set images for backdoor attack and the whole 10,000 testing-set images for defense. In the defense part, we randomly pick 80% images (8,000 images) to search the trigger's distribution using our algorithm. We use the remaining 20% images (2,000 images) to test our defense results. All of the accuracy results reported in the paper are evaluated on the testing set. Both the training set and the testing set are normalized to have zero mean and variance one. All the experiment are done using a single TITAN RTX GPU. The original Cifar10 model is a ResNet-18 model [21] build following `https://github.com/kuangliu/pytorch-cifar`'s instruction and reaching 95.0% accuracy.

During attack, we use stochastic gradient descent (SGD) [23] with $momentum = 0.9$, $learning\ rate = 0.001$ [24]to fine-tune the original model 10 more epochs with $batch\ size = 128$ and $poison\ ratio = 0.01$ for all of the 51 triggers and generate 51 backdoored models.

The step by step defense experiment includes:

1) Search distribution. The generator is a three-layer perceptron with $input\ dimension = 64$, $hidden\ layer = 512$. A batch normalization layer and a Leaky ReLu layer ($negative\ slope = 0.2$) are followed by both input layer and hidden layer. The input noise follows uniform distribution. The $output\ layer = 27$ is used to generate a $3 \times 3$ pixels with 3 channels trigger's pattern. Together with Generator, we have a Statistic Network to estimate entropy. The Statistic Network has two inputs. One input is the output of Generator, the other input is a uniform distribution noise (keep same with the Generator's input distribution). Each of them goes through a linear layer and transforms to a vector with 512 dimensions. Then the Statistic Network adds them together (with bias), following by two linear layers ($512 \times 512, 512 \times 1$), and generates an output. Each linear layer in Statistic Network follows with a leaky ReLu layer ($negative\ slope = 0.2$). When searching the backdoor distribution, we always train our Generator and Statistic Network 50 epochs using Adam optimizer [25] with $learning\ rate = 0.0002$ .

2) Use our generator to defend the backdoor attack. We defend the backdoored model through re-training using the same parameter as we attack it (SGD, $momentum = 0.9$, $learning\ rate = 0.001$, 10 epochs, $batch\ size = 128$ and $poison\ ratio = 0.01$). We apply model ensemble defense using all of the three models with $\beta = 0.9, 0.8, 0.5$. During 10 rounds defense, we sample the three models 4, 3, 3, times respectively to generate 10 different triggers in total .

3) Compare with baseline and ideal case. Baseline follows pixel space optimization. The initial trigger's pattern is generated from uniform distribution. Using cross entropy loss, SGD optimizer, $momentum = 0.9$, $learning\ rate = 0.1$, we search the trigger's pattern for 5 epochs. For each backdoored model, we search 10 patterns totally and calculate their mean and standard deviation of defense performance as our baseline. The ideal case uses the original trigger's pattern to perform defense, so it does not require extra search parameters. The defence parameters for both baseline and ideal cases are the same as the parameters that are used in attack (SGD, $momentum = 0.9$, $learning\ rate = 0.001$, 10 epochs, $batch\ size = 128$ and $poison\ ratio = 0.01$).

When plotting Figure. 5, we sample 2000 points from our Generator, and using t-SNE provided by sckit-learn [26] (init='pca', random_state=0). When plotting Figure. 3, we sample 128 points from our Generator and using PCA provided by sckit-learn (decomposition.TruncatedSVD). The minimum distance is set to $6e-3$ so as to prevent results from overlap.

**Supplementary experiments** To further test the performance of our method on a larger dataset, with different color triggers, and different target class, we keep the experimental environment as close to the previous one as possible. The differences between supplementary experiments and the main experiments are:

- Instead of always attacking class 0 on Cifar10, we randomly select 10 classes to attack on Cifar100
- Instead of using only black and white triggers, we generate 10 triggers with independent and uniform RGB colors on both Cifar10 and Cifar100
- Instead of setting the generator's $hidden\ layer = 512$ on Cifar10, we set $hidden\ layer = 2048$ on Cifar100.
- The original Cifar100 model is a ResNet-18 model build following `https://github.com/weiaicunzai/pytorch-cifar100`'s instruction and reaching 76.23% accuracy.

We keep other settings unchanged.