[Reviews · NeurIPS 2019]

Reviewer 1



Overall, I think this paper presents a valuable idea toward backdoor defending. The basic idea is to recover the triggers, while the main technical challenge that it solves is that different triggers being recovered may lead to poor defense performance. This work thus, instead of recovering one trigger, tries to model a distribution of the triggers. The technical development in Sec 3 makes sense, and the evaluation in Sec 4 demonstrates that the approach is valid. There are also several aspects that can be improved. The presentation (esp. the notations) in Sec 3 is very dense and hard to follow. For example, $g$ is a function, but I'm not sure what $g'$ is, and why it can be compared with a scalar $\omega$. Are $g$ and $g'$ just two linear transformation? The evaluation is done over CIFAR-10, while the triggers are all black/white. This may have some problem, since black-white triggers are clearly out of the distribution. The target class is set to c=0, and the authors argue that all classes are equivalent. But actually, they are not. It is natural to expect that some classes are easier to attack than others, as is in the case of adversarial example studies. Nevertheless, I think this is an OK paper, and the results are worth to be published.

Reviewer 2



-- The proposed method is novel and has some nice properties. It recovers the distribution of triggers, thereby brings better robustness to the network. -- The experiments are comprehensive and demonstrate the effectiveness and robustness of the model. -- This work is well-written with barely grammar mistakes. -- Figure 1 can be reorganized for better clarity. -- I do not see major weaknesses in the work. The experiments on more datasets like CIFAR100 and Tiny ImageNet could be a plus.

Reviewer 3



Originality: This paper contains several points of novelty: discovery of the continuous nature of trigger distributions, defining a new generative procedure -- MESA -- from an unknown distribution without samples, and applying MESA to defense against backdoor triggers to achieve significantly improved performance. Quality: The proposed approach is principled and can effectively learn an unknown distribution without samples when given enough computational resources. The assumption that the testing function F is the composition of the density f and a monotonic function g is somewhat idealistic, but this assumption seems to hold reasonably well for the task for backdoor trigger detection. Clarity: This is the main area of weakness for this paper. The description of MESA (Section 3.1) is very messy and unstructured, and it took several passes for me to understand the method. In particular, I could not fully appreciate why the ensemble was necessary until realizing that each G_i is learning a level set of the true density function. To help the reader understand and appreciate the result, the authors should devote significant effort to clearly explaining the method, e.g. via a step-by-step breakdown of Figure 2(a). Another minor question is regarding the experimental setting. When randomly sampling the location to apply the trigger, is the location fixed for all triggers or is it also variable? In this case, does F need to randomly apply the trigger to a chosen location and repeat several times to compute the average success rate across locations? If not, I see this as a flaw in the evaluation setting as it is unreasonable to assume that the defender knows the (random) location of the trigger. Significance: The improvement that MESA makes over prior defense is significant as demonstrated by Figure 4. This work makes several important discoveries central to the testing-stage defense of backdoor trigger attacks. I can foresee an array of future work that proceeds in this direction. One clear limitation of this method is that the support set of the distribution that MESA models must be small. In particular, the authors limit to 51 patterns in their evaluation as opposed to all 3x3x3 (CxHxW) tensors containing RGB values between 0 and 255. This is reasonable as a first step, but I am concerned that this approach is limited in principle by the size of the support set in that the computation may scale linearly with the number of supported elements. ---------------------------------------------------------------------- I have read the author response and decided to maintain my evaluation.

Reviewer 4



(1) This paper is clearly written. Useful visualizations (e.g., Figure 3 & 5) are provided for better understanding of this paper. (2) This paper is well motivated --- by empirically show that there indeed exists a distribution for backdoor triggers, it then makes a perfect sense that we should apply a generative model to characterize this distribution for obtaining strong defense model. (3) The proposed method is novel. As the backdoor trigger distribution is unknown, traditional generative modeling methods, like GAN or VAE, which need directly sampling from the distribution is not applicable in this situation. Therefore, the author proposes a sampling-free algorithm (which based on entropy maximization) to model the trigger distribution. (4) Compared to traditional reverse engineering method, experiment results on CIFAR-10 demonstrate that the proposed method can improve and stabilize the robustness of defense models.

[Author Response · NeurIPS 2019]

**Clarification of the MESA method (Review #2, #4).** We are aware that the dense notations in Section 3.1 is hard to follow. We will rewrite the part in a more descriptive language and move the excessive details to Appendix. For the question by Reviewer #2, $g'$ is the derivative of $g$. The $g' > \omega$ in line 149 should be corrected as $g'(\beta) > \omega, \forall \beta$.

**The choice of dataset, target class, and black-white trigger (Review #2, #3, #4).** Unlike adversarial attacks that are closely related to the dataset (especially the decision boundary), backdoor attacks inject arbitrary triggers that have little relationship with the dataset. The trigger is usually deliberately designed to

Figure 1: We generate 10 triggers with independent and uniform RGB colors, and test them on CIFAR-10/CIFAR-100 with target class 0/random.

be out-of-distribution to achieve a high attack success rate. From the feature space's perspective, we can assume that normal data from all classes form one cluster, and poisoned data form another cluster. Previous research [1] explicitly used this property for backdoor defense. The intuition is that the distance between normal data and poisoned data, instead of the dataset and class labels, determines the attack/defense difficulty.

Without worrying about the out-of-distribution issue, we chose black-white triggers for a better coverage of corner cases. Naively randomizing RGB colors with [0, 255] values can (almost) never generate special triggers such as a 3×3 black square (requiring 27 zeros). We add several experiments using random-color triggers as shown in Figure 1. Our defense method obtains ∼2% after-defense ASR (attack success rate) in average, better than the previous results on black-white triggers (∼4%). The experiments are performed on both CIFAR-10 (Figure 1(a), fixed target class) and CIFAR-100 (Figure 1(b), random target class) to show the marginal effect of dataset and target class choices. In the final submission, we will include the discussions on the impact of trigger color, dataset, and target class.

Regarding to Reviewer #4's concern about the size of the support set, the choice of black-white and colorful triggers only decides the support set of original triggers, not reversed triggers. The search space of reverse engineering is always $\mathbb{R}^{27}$ regardless of the choice of original triggers.

**Trigger locations (Review #4).** We randomize the trigger location in each attacking step, and the backdoored model is sensitive to the trigger at any location. The defense algorithm tests a generated trigger at a random location in each step, and trains the generator with gradients from all locations. The only prior knowledge is the 3×3 trigger size.

**Comparing to related works about model ensembling (Review #5).** GWN [2] trains multiple models towards the same target distribution and introduces inter-model interaction to improve image diversity. DoPaNet [3] trains multiple discriminators targeting different modes to reduce the modeling complexity. Both methods use model ensembling as an enhancement to the original GAN and solve the mode dropping problem. Without model ensembling, a single GAN can still theoretically model any arbitrary distribution, if ignoring the model capacity limitation in practice.

The model ensembling in this work has a completely different motivation. In the sampling-free setting, the Nash equilibrium in GANs does not exist, and we have to train the generator without any discriminator. Under this constraint, we find that directly learning an arbitrary distribution being difficult, and simplify the problem by targeting uniform distributions. Then, model ensembling becomes a mandatory step to recover the arbitrary distribution from uniform distributions. A single model is incapable to capture the arbitrary distribution both theoretically and practically.

**Advantage of model ensembling and justification of parameter selection (Review #5).** The optimal backdoor defense is to retrain the backdoored model using the original trigger. Reverse engineering the trigger distribution is only an alternative approach to cover the original trigger. The ensemble model does not necessarily perform better than a single sub-model, since any sub-model has a chance to cover the original trigger.

Model ensembling mainly serves for two purposes: 1) use multiple cross-sections to allow us to better understand the shape of trigger distribution and 2) provide a robust defense without parameter tuning. Parameter $\alpha$ balances the hinge loss and regularization. Fixing $\alpha$ to any value between 0.1 and 1 is empirically okay. Parameter $\beta$ is related to the strength of the attack and an inappropriate value leads to less effective defense (the reversed trigger distribution can be too sparse or too narrow to cover the original trigger). When the attack is unknown and $\beta$ cannot be predetermined, an ensemble model provides a robust defense without tuning $\beta$ (see yellow lines in Figure 1).

[1] Brandon Tran et al., Spectral Signatures in Backdoor Attacks, In NeurIPS, 2018.

[2] Honglun Zhang et al., Generative Warfare Nets: Ensemble via Adversaries and Collaborators, In IJCAI, 2018.

[3] Botos Csaba et al., Domain Partitioning Network, arXiv:1902.08134, 2019.


[Meta-Review · NeurIPS 2019]

Thanks for the paper submission and for addressing concerns in the reviews. We believe this is a valuable contribution to NeurIPS. The main contribution on backdoor defenses for neural networks is interesting and may open the door to further research in this direction. As areas of improvement, we strongly recommend simplifying the densest parts of the presentation (as long as completeness is maintained), enhance the related work (as addressed in the author feedback), and class equivalency (similarly trigger choice). Please refer to the detailed feedback for valuable suggestions and comments from the reviewers.